# Chemical Characterization and Anti-HIV-1 Activity Assessment of Iridoids and Flavonols from *Scrophularia trifoliata*

**DOI:** 10.3390/molecules26164777

**Published:** 2021-08-06

**Authors:** Francesca Guzzo, Rosita Russo, Cinzia Sanna, Odeta Celaj, Alessia Caredda, Angela Corona, Enzo Tramontano, Antonio Fiorentino, Francesca Esposito, Brigida D’Abrosca

**Affiliations:** 1Department of Environmental Biological and Pharmaceutical Sciences and Technologies, DiSTABiF University of Campania Luigi Vanvitelli, Via Vivaldi 43, 81100 Caserta, Italy; francesca.guzzo@unicampania.it (F.G.); rosita.russo@unicampania.it (R.R.); odeta.celaj@unicampania.it (O.C.); antonio.fiorentino@unicampania.it (A.F.); 2Department of Life and Environmental Sciences, University of Cagliari, Via Sant’Ignazio da Laconi 13, 09123 Cagliari, Italy; cinziasanna@unica.it; 3Department of Life and Environmental Sciences, University of Cagliari, Cittadella Universitaria di Monserrato, ss554, km 4500, Monserrato, 09042 Cagliari, Italy; alessiacaredda@unica.it (A.C.); angela.corona@unica.it (A.C.); tramon@unica.it (E.T.); 4Department of Marine Biotechnologies, Stazione Zoologica Anton Dohrn, Villa Comunale, 80121 Naples, Italy

**Keywords:** *Scrophularia trifoliata*, iridoids, flavonoids, NMR, structural characterization, HIV-1 integrase, RT-associated RNase H

## Abstract

Plants are the everlasting source of a wide spectrum of specialized metabolites, characterized by wide variability in term of chemical structures and different biological properties such antiviral activity. In the search for novel antiviral agents against Human Immunodeficiency Virus type 1 (HIV-1) from plants, the phytochemical investigation of *Scrophularia trifoliata* L. led us to isolate and characterize four flavonols glycosides along with nine iridoid glycosides, two of them, **5** and **13**, described for the first time. In the present study, we investigated, for the first time, the contents of a methanol extract of *S. trifoliata* leaves, in order to explore the potential antiviral activity against HIV-1. The antiviral activity was evaluated in biochemical assays for the inhibition of HIV-1Reverse Transcriptase (RT)-associated Ribonuclease H (RNase H) activity and HIV-1 Integrase (IN). Three isolated flavonoids, rutin, kaempferol-7-*O*-rhamnosyl-3-*O*-glucopyranoside, and kaempferol-3-*O*-glucopyranoside, **8**–**10**, inhibited specifically the HIV-1 IN activity at submicromolar concentration, with the latter being the most potent, showing an IC_50_ value of 24 nM.

## 1. Introduction

*Scrophularia trifoliata* L., belonging to the family of Scrophulariaceae, is an endemic plant of Sardinia, Corsica, and Gorgona Islands [1]. It is an herbaceous perennial plant, with a woody base, growing up to 1.5–2.0 m height. It is found predominantly in fresh and shady places. The plant is characterized by opposite and lanceolate leaves and tetragon, fistulous multistems. The irregular flowers have a green or red bilabiate corolla, which has the unique characteristic of two reddish or reddish-purple spots, surrounded by a broad black stripe [2]. These features act as nectar guides for animals and insect pollinators, involved in the so called “*bird and mixed vertebrate–insect pollination system*” (MVI); in addition, their peculiar colorations are indicative of a high anthocyanin content [3].

The plant is included in the large genus *Scrophularia*, which comprises about 60 species from where different classes of specialized metabolites have been isolated. Among them, iridoids and flavonoids have proven to produce several biological activities [4,5].

For example, scropolioside B isolated from *S. dentata* “Ye-Xin-Ban” had an inhibitory effect against nuclear factor kappa-light-chain-enhancer of activated B cells [6]; harpagide from *S. buergeriana* showed a good protective effect against glutamate-induced oxidative stress on cultured neurons [7]; ajugoside, isolated from *S. deserti*, exhibited antibacterial activity against multidrug and methicillin-resistant *S. aureus* [8]. Moreover, flavonoids such as nepitrin, acacetin, and scrophulein are reported as constituents of *S. striata* [9], *S. takesimensis* [10], and *S. grossheimii* [11], respectively, showing many bioactivities such as antioxidant, antibacterial, anti-inflammatory, and anti-nociceptive [12,13]. Some ethnobotanical studies reveal the traditional medical use of *S. trifoliata* to treat many diseases. To mention a few, poultice of fresh leaves with olive oil is used in Sardinia against rheumatism [14], while infusion of leaves and poultice of flowers are used to treat skin disorders [15].

These findings motivated the investigation of *S. trifoliata*, based on two interesting considerations. Despite its acceptance and documented uses as antinflammatory, antirheumatic, diuretic, vulnerary, and anthelmintic among Sardinian inhabitants, it is a species underexplored [16], indeed, only one phytochemical study has been reported until now [17]. However, overall *S. trifoliata* is a plant of endemic Sardinian flora and some of Sardinian endemism have shown peculiar specialized metabolites patterns [18]. Such chemical diversity could be due to geographical isolation, which generates genetic and metabolic differentiation [16]. In fact, chemical composition is peculiar for Sardinian vegetation in respect to the plants growing in the Italian peninsula and some of Sardinian endemism plants highlighted also very interesting biological and pharmacological activities as already documented in literature data [19,20].

It is worth noting that in our previous investigation on Sardinian endemic flora, we found some species endowed with anti-HIV-1 activity [20,21] and in some cases the antiviral activity was due to compounds peculiar to these plants [19].

In the present study, we investigated, for the first time, the contents of a methanol extract of *S. trifoliata* leaves, in order to explore their potential antiviral activity against Human Immunodeficiency Virus type 1 (HIV-1). In particular, the phytochemical investigation led to isolate nine iridoids, two of them isolated for the first time, and four flavonols glycosides from bioactive fractions. The antiviral activity has been evaluated in biochemical assays for the inhibition of HIV-1 Reverse Transcriptase (RT)-associated Ribonuclease H (RNase H) activity, and for the inhibition of HIV-1 integrase (IN) in the presence of the LEDGF7p75 cellular cofactor.

Results showed that three isolated compounds specifically inhibited HIV-1 IN, being inactive on HIV-1 RNase H, at sub-micromolar concentration. Among the three, kaempferol-3-*O*-glucopyranoside was the most potent, showing an IC_50_ value of 24 nM.

## 2. Results

### 2.1. Anti-HIV Activity of Methanolic Crude Extract

With the aim to find new agents that dual inhibit HIV-1 RNase H and IN activities, we tested the crude extract of *S. trifoliata*, firstly for its ability to inhibit the RT-associated RNase H activity, for which no drug is currently available. Then, it was assayed for its effects also on HIV-1 IN in biochemical assays. Results were expressed as IC_50_ values against HIV-1 RNase H activity and in IN LEDGF-dependent integration. The extract showed an evident inhibitory activity with IC_50_ values of 9.90 ± 0.93 and 2.5 ± 0.4 μg/mL on HIV-1 RNase H activity and HIV-1 IN strand transfer activity, respectively.

### 2.2. 2D-NMR Investigation of Crude Extract

In the attempt to characterize the metabolites potentially responsible for highlighted anti-HIV activities, an extensive 2D-NMR study of the crude extract was carried out. The ^1^H-NMR spectrum of *S. trifoliata* (Figure 1) was dominated by resonance of iridoids with a characteristic doublet at δ_H_ 5.04 as well as two double doublets at 6.38 (dd, *J* = 1.2 and 5.7 Hz) and 5.08 (dd, *J* = 4.2 and 5.7 Hz) related to dihydropyrane ring. Several overlapped signals were also detected in the region of proton germinal to oxygen as well as in the up-field region of ^1^H-NMR spectrum.

The doublet of doublets at δ_H_ 6.38 (δ_C_ 141.4) showed long range heterocorrelation in the CIGAR-HMBC with a methine carbon at δ_C_ 36.7 (δ_H_ 2.33), an olefin carbon at δ_C_ 103.7 (δ_H_ 5.08), and an acetalic carbon at δ_C_ 95.1 bonded to a doublet resonating at δ_H_ 5.04. This signal was, in turn, correlated with three methine at δ_C_ 141.4, δ_C_ 34.7, and finally an anomeric carbon at δ_C_ 99.3. This latter suggested the presence of a glycosidic moiety in this compound, identified as glucose based on spectroscopic evidence. The ^13^C-NMR values at 99.3, 77.4, 77.1, 74.4, 71.0, and 62.1 and the coupling constant value of 8.1 Hz are in good agreement with the β-glucopyranosil moiety. The acetalic proton at δ_H_ 5.04 showed correlations, in the HSQC-TOCSY experiment, with two carbinolic carbon at δ_C_ 95.1 and δ_C_ 87.4, and two aliphatic methine at δ_C_ 42.5 and 36.7. This latter, in the same experiment (Figure 2), showed correlations with all the protons belonging to the same spin system: 6.38 (H-3), 5.08 (H-4), 2,58 (H-9), 2.33 (H-5), and 3.74 (H-6). This latter proton, linked to methine carbon at 87.4, showed long range correlations, in the CIGAR-HMBC, also with a proton at 3.77, linked to carbon at 58.0 in good accordance with the presence of an oxirane ring. Thanks to long range correlations of both carbons of epoxide (δ_C_ at 58.0 and 66.0) with H-9 and methylene proton of a hydroxyethyl at 4.20 and 3.80, the oxyrane ring is located at the C-7 and C-8 carbons of iridoid skeleton.

All data were in agreement with those reported for catalpol (**1**, Figure 1), already reported as a constituent of *S. trifoliata* [17].

Moreover, it was evident the presence of other signals belonging to other iridoids glycosides, which were present in low quantities, and other signals in the range of 6.0–8.0 ppm, attributable to flavonoids and other metabolites. Nevertheless, the identification of compounds in the mixture was not possible, hence, the extract was subjected to phytochemical study in order to identify the compounds responsible for the HIV-1 RNase H and IN activities.

### 2.3. Phytochemical Study of Scrophularia trifoliata

The crude methanolic extract of *S. trifoliata* was purified on Amberlite XAD-4, first with water and then eluting with MeOH, in order to eliminate sugar and other water-soluble metabolites. The alcoholic eluate then was fractionated through column chromatography RP-18, obtaining seven fractions, A–G (Figure 3). Fractions were preliminary analyzed by thin-layer chromatographic (TLC) plate, eluting with the lower phase of CHCl_3_/MeOH/H_2_O (13:7:2) solution. Fraction G showed a more complex TLC profile and the NMR spectrum suggested the presence of flavonoids and iridoids.

The combination of different chromatographic procedures led to isolate, from fraction A–F, a new iridoid glycoside (**5**), along with another six already known compounds (**1**–**4**, **6**–**7**).

The MALDI mass spectrum of compound **5** showed quasimolecular peaks of the adducts at *m*/*z* 399.48 [M + Na]^+^, *m*/*z* 415.46 [M + K]^+^, and *m*/*z* 377.71 [M + H]^+^ according to a molecular formula C_16_H_24_O_10_. The ^1^H-NMR spectrum (Table 1) showed signals related two acetalic hydrogens at δ_H_ 5.62 (d, *J* = 4.2 Hz) and 4.65 (d, *J* = 7.8 Hz) ppm, two olefinic protons at δ_H_ 6.28 (dd, *J* = 6.3, 1.8 Hz) and 5.03 (dd, *J* = 6.3, 3.6 Hz) ppm, and, in the down-field region, signals of two hydrogens at δ_H_ 2.45 (dd, *J* = 10.5 Hz) and a multiplet at δ_H_ 2.77.

The ^13^C-NMR showed 16 carbons identified, based on the HSQC experiment (Figure 4), as 1 methoxyl (δ_C_ 57.5), 2 methylene carbons, 12 methines, and 1 tetrasubstituted carbon at δ_C_ 81.5. The signals at δ_C_ 99.6 (C-1′), δ_C_ 74.8 (C-2′), δ_C_ 77.8 (C-3′), δ_C_ 71.6 (C-4′), δ_C_ 77.9 (C-5′), and δ_C_ 63.7 (C-6′) correlated to protons of the same spin system (HSQC-TOCSY), suggesting the presence of a glucose moiety.

The coupling constant of the anomeric proton (7.8 Hz) suggested a β-configuration for the anomeric carbon.

In addition, the CIGAR-HMBC (Appendix A Appendix A) correlations between δ_H_ 6.30 (H3) and δ_C_ 93.0 (C-1), between olefinic proton H4 at δ_H_ 5.03 and carbons at δ_C_ 140.3 (C-3), δ_C_ 37.1 (C5), δ_C_ 93.5 (C-6), and δ_C_ 48.9 (C-9) were evident. In the same experiment, long-range correlations of acetalic proton H1 (δ_H_ 5.62) and δ_C_ 81.5 (C-8) and δ_C_ 48.9(C-9) were detected.

Furthermore, the H-10 methylene protons correlated, in turn, with C-7, C-8, and C-9 carbons, which supported the presence of one 7,10-oxetane group in compound 5.

The downfield shift of the C-6 carbon at δ_C_ 93.5 with respect to the same signal of cymdahoside A (δ_C_ 77.6) [22] suggested the linkage of a methoxy group to C-6 carbon (Table 1). The heterocorrelation in the CIGAR-HMBC experiment between the singlet methyl at δ_H_ 3.46 and the carbon at δ_C_ 93.5 (C-6) confirmed this hypothesis. Similar differences were registered between methylcatalpol (**4**) and catalpol (**1**). Finally, the correlations observed between anomeric proton at δ_H_ 4.65 (H-1′) and carbon at δ_C_ 91.6 (C-1), and vice versa in the CIGAR-HMBC spectra, allowed us to link a β-glucopyranosyl unit of a C-1 of iridoid structure.

The stereochemistry of ring fusion was confirmed by the coupling constant *J* = 10.2 Hz between crumpled protons H9 (δ_H_ 2.45) and H-5 (δ_H_ 2.77). The NOESY (Appendix A Appendix A) experiment enabled the relative configuration at the chiral carbons to be defined. In fact, the NOE observed between the β-oriented H-9 proton and H-5 and H-7 protons agreed with a β-orientation of both C-7 and C-5 methine, corresponding to R configuration for the C-5, C-7, and C-9 carbons.

Thus, compound **5** (Figure 5) was characterized for the first time and named trifoliatoside A.

The known iridoids were identified as catalpol (**1**) and aucubin (**2**) [23], ajugol (**3**) [24], methylcatalpol (**4**) [25], methylscutelloside (**6**) [26], and ajugoside (**7**) [27], based on a comparison between their spectroscopic features and the literature data.

### 2.4. Anti-HIV Activity of Pure Compounds and Fraction G

Pure metabolites **1**–**7** and fraction G were evaluated for their anti-HIV properties in enzymatic assays. Among isolated metabolites, only compounds **5** and **7** were able to inhibit IN, selectively, even though at high concentration (Table 2), while the enriched fraction was active on both RNase H and IN activities with IC_50_ values of 26.6 ± 1.3 µg/mL and 6.1 ± 0.92 µg/mL, respectively. Even though the concentrations were higher than those tested for the alcoholic extract, these results brought us to a phytochemical study of enriched fraction, trying to identify the pure compounds responsible for this effect.

### 2.5. Phytochemical Investigation of Fraction Active G: Structural Characterization of Flavonols and Acylated Iridoid Glicosides

Based on the observed anti-HIV activity, fraction G was further chromatographed on column chromatography RP-18, furnishing four fractions, G1–G4. The NMR profiling of these fractions suggested the presence of flavonoids and iridoids. In fact, the known flavonoids, rutin (**8**) [28], kaempferol-7-*O*-rhamnosyl-3-*O*-glucopyranoside (**9**) [29], kaempferol-3-*O*-rutinoside (**10**) [30], and kaempferol-3-*O*-glucopyranoside (**11**) [28] were identified as constituents of fraction G (Figure 3), along with two acylated iridoid glycosides (**12**–**13**, Figure 5), one of them (**13**) isolated and characterized for the first time.

Compound **13** (Table 3) showed a molecular formula C_24_H_30_O_10_, as deduced from the elemental analysis and the MALDI mass spectrum, which showed the quasimolecular peaks at *m*/*z* 478.65 [M + H]^+^, *m*/*z* 501.40 [M + Na]^+^, and 517.38 [M + K]^+^. The ^13^C-NMR showed 24 carbons including 1 methyl carbon, 2 methylenes, 18 methines, and 2 quaternary carbons; one of them suggested the presence of an ester group according to chemical shift value at δ_C_ 168.2.

The ^1^H-NMR spectrum of compound **13** was dominated by caracteristis resonances related to the dihydropyrane ring of iridoid (Table 3) besides two multiplets at δ_H_ 4.00 (δ_C_ 76.8) and δ_H_ 2.14 (δ_C_ 48.5). These latter methylene protons correlated, in the COSY spectrum, with a proton at δ_H_ 4.00 (H-6), which in turn showed cross peak with H-5 (δ_H_ 2.80). Others homocorrelations highlighted in the COSY spectrums (Figure 6) supported by HSQC experiment (Table 3) are in good agreement with the presence of ajugol as aglycon (Figure 5). The heterocorrelation present in HMBC between C-1 of ajugol [24] with anomeric proton of glucose unit suggested C-1 as the glycosylation site.

However, the presence in the aromatic region of the ^1^H-NMR spectrum of additional signals along with HSQC data suggested the presence of a cinnamoyl moiety. The coupling constant value of 12.6 Hz for olefinic protons H-7″ (δ_C_ 142.4) and H-8″ (δ_C_ 122.3) is in good agreement with a Z-geometry for a double bond of cinnamoyl moiety linked to C-8 of iridoid. In fact, in the HMBC experiment, the proton of methyl group (δ_H_ 1.58) showed heterocorrelation with C-8″ (δ_C_ 122.3), C-9 (δ_C_ 49.4), and C-8 (δ_C_ 90.3). Furthermore, the down-field shift of C-8 chemical shift that bears cinnamoyl unit is in good agreement with the presence of an acyl moiety.

Thus, the structure of compound **13** was elucidated and named as trifoliatoside B. Also, the laterioside, **12**, [31], was from leaves of *S. trifoliata*.

### 2.6. Anti-HIV Activity of Pure Compounds Isolated from Fraction G

Pure compounds exhibited selective inhibition properties against IN-LEDGF-dependent activity (Table 4). In particular, the most active were compounds **11** and **9**, which showed IC_50_ values of 0.11 and 0.024 µM, respectively. In addition, **8**, **10**, and the new compound **13** inhibited the HIV-1 IN in an IC_50_ values range between 0.33 and 5.96 µM. Differently, compound **12** was found inactive on both enzymes’ activities.

It was interesting to note that compounds **9**, **10**, and **11**, even though sharing the same aglycon moiety, exhibited different activities. In fact, the best activity was reported for **11**, which presents a minor grade of glycosylation in respect to the other flavonoids tested, which are linked at two sugar units. In addition, we can hypothesize that this compound has two different modes of action against the HIV-1 virus, based on the fact that anti-HIV-1 protease activity is already reported in the literature [32]. This is the first report about the HIV-1 IN-LEDGF-dependent inhibition of compound **13**.

## 3. Discussion

*Scrophularia trifoliata* L. is an endemic plant of Sardinia, which is known to be used in traditional medical practices of the island for the treatment of different diseases, such as skin disorders and rheumatism [16]. Despite these traditional uses, to the best of our knowledge, to date only one phytochemical study on *S. trifoliata* has been reported [17]. This study that analyzed the monoterpenoid fractions led to isolate two C_9_ iridod glycosides: catalpol and aucubin, which were also detected in the present investigation. Aucubin, bartioside, harpagoside from *S. scorodonia* leaves [33], 6-*O*-methylcatalpol, harpagide from *S. ningpoensis* roots [34] and 8-*O*-acetylharpagide, and scropolioside B from *S. saharae* [35] are just example of iridoids that have been isolated over the years from a little percent (4%) of investigated *Scropularia* species [4]. Recently, Venditti et al., 2015 [36] underlined some glycosidic iridoids also from *S. canina* living in Calabria region (Italy). So, C_9_ iridoids in glycosidic and non-glycosidic forms could be considered chemotaxonomy markers of Scrophulariaceae family [4,37]. In this way, the present study furnishes important information in term of phytochemical composition of another endemic plant of Sardinia (Italy), highlighting the presence of iridoid glycosides also in *S. trifoliata*. Furthermore, this study contributed to increase the percentage of investigated Scrophulariaceae species [4] against the total of uninvestigated (only 17 of the approx. 350 species). Besides iridoids, also flavonoids and phenilethanoid glycosides have been isolated from different species belonging to *Scrophularia* genus [4].

All these compounds amply studied for their antioxidant, antibacterial, and anti-inflammatory activities have not been sufficiently investigated for antiviral properties [33,38].

Furthermore, in this study, we tested *S. trifoliata* pure compounds on two HIV-1 viral enzymes: HIV-1 Reverse Transcriptase (RT)-associated Ribonuclease H (RNase H) activity, a promising target for which no drug is currently available in the therapy [39,40,41], and for the inhibition of HIV-1 integrase (IN) in the presence of the LEDGF7p75 cellular cofactor [42], a protein that is able to bind the IN to promote its catalytic activities, trying to identify potential anti-HIV therapeutic agents. In fact, HIV-1 RNase H and IN activities are viral-encoded enzymes that belong to the nucleotidil transferase superfamily and, therefore, possess homologies in their structure; in this contest, we discovered N’-acylhydrazones that inhibit one or both HIV-1 RT-associated RNase H activity and IN enzyme activities [43].

Today, the research of anti-HIV 1 agents is looking for plant-derived compounds, which can be an essential natural source of antiretroviral with low toxicity and a wide spectrum of actions. Out of them, flavonoids were reported for their ability to inhibit HIV-1 IN [39]. In addition, kaempferol from *Securigera securidaca* [38] and baicalin from *Scutellaria baicalensis* [44] inhibited the HIV-1 RT with IC_50_ values of 50 µg/mL and 0.2 µg/mL, respectively; taxifolin from *Juglans mandshurica* inhibited the enzymes protease and RT [45]. On the contrary, only the iridoid glycoside 2′-*O*-(4-methoxycinnamoyl) mussaenosidic acid, isolated from *Avicenna marina*, has been reported for its ability to prevent viral infection, acting on co-receptors CCR5 and CXCR4 [38]. In this study, a bio-guided phytochemical approach led us to purification of iridoid glycosides and flavonoid compounds from methanol extract of *S. trifoliata* leaves, some of which showed a potential anti-HIV activity. In fact, the newly identified iridoid glycoside cis-laterioside and all isolated flavonoids exhibited a significant inhibition of integrase. We plan to complete in the future the study of the main active compounds with docking molecular analysis, providing some information about their interaction mechanism, and to proper pharmaceutical formulations for medical applications.

## 4. Materials and Methods

### 4.1. Plant Material

Leaves of *S. trifoliata* were collected at the flowering stage (April 2016) in the site of Seui (Sardinia, Italy, 39°49′52.3″ N, 9°20′31″ E-693 m a.s.l.). The plant was identified and a voucher specimen (Herbarium CAG 1011) was deposited at the General Herbarium of the Department of Life and Environmental Sciences, University of Cagliari (Cagliari, Italy). *S. trifoliata*, even if endemic, is not protected by local or international regulations, therefore, no specific permission was required for its collection. The plant raw materials were dried in a ventilated stove at 40 °C to constant weight, powdered with liquid nitrogen, and stored at −20 °C until next analysis.

### 4.2. Preparation of Crude Extract for Bioassay

The powder obtained from the leaves of *S. trifoliata* (427 g) was extracted with MeOH (3 × 1 L, 8 h each), filtered, and evaporated to obtain crude extract (22 g) that was submitted for biological evaluation.

### 4.3. Extraction Procedure for NMR Analysis

Powdered air-dried leaf material of *S. trifoliata* (600 mg) underwent ultrasound assisted extraction (Branson 3800 MH, Milan, IT), with a H_2_O/MeOH (1:1) solution (18 mL), for 40 min. Subsequently, the mixture was centrifuged (Beckman Coulter’s AllegraTM 64R centrifuge; rotore F1202, r = 3.5 cm) a 13,000 rpm, for 10 min. After filtration and solvent removal, a crude extract was obtained and stored at −20 °C until analyses. To obtain enriched fractions in specialized metabolites, potentially responsible for biological activities, the crude extract was filtered through a Sep-Pak C18 classic short body cartridge (Waters, Milford MA, USA), previously conditioned with MeOH (10 mL), followed by H_2_O (10 mL). The resulting residue was dissolved in a volume of 1.5 mL of phosphate buffer in D_2_O and methanol-d4 (1:1). An aliquot of 0.6 mL was transferred to an NMR tube and analyzed by NMR [46].

### 4.4. General Chromatographic Procedures

Analytical TLC was performed on Merck Kieselgel 60 F254 or RP-8 F254 plates of 0.2 mm layer thickness. The plate was visualized by UV light or by spraying with H_2_SO_4_/AcOH/H_2_O (1:20:4), followed by heating at 120 °C for about 1 min. The plates were then heated for 5 min at 120 °C. Preparative TLC was performed on Merck Kieselgel 60 F254 plates, of 0.5 or 1 mm film thickness. Column chromatography (CC) was performed on Fluka (Seelze, Germany) Amberlite XAD-4, on Merck Kieselgel 60 (70–240 µm) and Merck Kieselgel 60 (40–63 µm), or on Baker (Deventer, The Netherlands) RP-18.

### 4.5. NMR Experiments

NMR spectra were recorded at 25 °C on 300.03 MHz for ^1^H and 75.45 MHz for ^13^C on a AVANCE II 300 MHz NMR Spectrometer Fourier transform in CD_3_OD or CD_3_OD: phosphate buffer solutions at 25 °C. Chemical shifts are reported in δ (ppm), and referenced to the residual solvent signal; *J* (coupling constant) are given in Hz.^1^H-NMR spectra were acquired over a spectral window from 14 to −2 ppm, with 1.0 s relaxation delay, 1.70 s acquisition time (AQ), and 90° pulse width = 13.8 μs. The initial matrix was zero-filled to 64 K. ^13^C-NMR spectra were recorded in ^1^H broadband decoupling mode, over a spectral window from 235 to −15 ppm, 1.5 s relaxation delay, 90° pulse width = 9.50 μs, and AQ = 0.9 s. The number of scans for both ^1^H and ^13^C-NMR experiments were chosen, depending on the concentration of the samples. With regards to the homonuclear and heteronuclear 2D-NMR experiments, the data points, number of scans, and increments were adjusted according to the sample concentrations. Correlation spectroscopy (COSY) and double quantum filtered COSY (DQF-COSY) spectra were recorded with gradient-enhanced sequence at spectral widths of 3000 Hz in both f2 and f1 domains; the relaxation delays were of 1.0 s. The total correlation spectroscopy (TOCSY) experiments were performed in the phase-sensitive mode with a mixing time of 90 ms. The spectral width was 3000 Hz. Nuclear Overhauser effect spectroscopy (NOESY) experiments were performed in the phase-sensitive mode. The mixing time was 500 ms, and the spectral width was 3000 Hz. For all the homonuclear experiments, the initial matrix of 512 × 512 data points was zero-filled to give a final matrix of 1 k × 1 k points. Proton-detected heteronuclear correlations were also measured. Heteronuclear single-quantum coherence (HSQC) experiments (optimized for ^1^*J* (H, C) = 140 Hz) were performed in the phase sensitive mode with field gradient. The spectral width was 12,000 Hz in f1 (^13^C) and 3000 Hz in f2 (1H), and had 1.0 s relaxation delay; the matrix of 1 k × 1 k data points was zero-filled to give a final matrix of 2 k × 2 k points. Heteronuclear 2 bond correlation (H2BC) spectra were obtained with T = 30.0 ms and a relaxation delay of 1.0 s; the third order low-pass filter was set for 130 < ^1^*J*_(C,H)_ < 165 Hz. A heteronuclear multiple bond coherence (HMBC) experiment (optimized for ^1^*J*_(H,C)_ = 8 Hz) was performed in the absolute value mode with field gradient; typically, ^1^H–^13^C gHMBC were acquired with spectral width of 18,000 Hz in f1 (^13^C) and 3000 Hz in f2 (^1^H) and 1.0 s of relaxation delay; the matrix of 1 k × 1 k data points was zero-filled to give a final matrix of 4 k × 4 k points. Constant time inverse-detected gradient accordion rescaled heteronuclear multiple bond correlation spectroscopy (CIGAR–HMBC) spectra (8 > ^n^J_(H,C)_ > 5) were acquired with the same spectral width used for HMBC. Heteronuclear single quantum coherence-total correlation spectroscopy (HSQC-TOCSY) experiments were optimized for ^n^*J*_(H, C)_ = 8 Hz, with a mixing time of 90 ms.

### 4.6. MALDI-TOF MS Analyses

Mass spectrometry analyses of pure compounds were performed with a matrix assisted laser desorption ionization time-of-flight (MALDI-TOF) mass spectrometer equipped with a pulsed nitrogen laser (λ = 337 nm). Prior to the acquisition of spectra, 1 μL of sample solution (100 pmol/μL) was mixed with 1 μL of saturated α-cyano-4-hydroxycinnamic acid matrix solution (10 mg/mL in acetonitrile/trifluoroacetic acid 0.1%, 1:1, *v*:*v*) and a droplet of the resulting mixture (1 μL) placed on the mass spectrometer’s sample target. The droplet was dried at room temperature. Once the liquid was completely evaporated, the sample was loaded into the mass spectrometer and analyzed in positive reflectron mode. The instrument was externally calibrated using a 50 fmol/μL tryptic alcohol dehydrogenase digest. The instrument source voltage was set at 12 kV.

### 4.7. Hydroalcoholic Extraction of S. trifoliata Leaves and Compounds Purification

Dried leaf material of *S. trifoliata* (40 g) was powdered and extracted by ultrasound assisted extraction (Branson 3800 MH) for 40 min each and three cycles with H_2_O/MeOH (1:1) solution (1.2 L). The flask was centrifuged. Subsequently, after centrifugation at 4800 rpm (Beckamn, GS-15R centrifughe; rotore S418; r = 3.5 cm) for 10 min at 22° C, the extract was filtered on Whatman paper and concentrated under vacuum, furnishing a dried crude extract (16.6 g).

The dried crude extract dissolved in H_2_O was purified on Amberlite XAD-4 column first with water in order to eliminate sugars and other water-soluble compounds. Successively, methanol elution furnished 1.2 g of residual material purified on RP-18 CC eluting with decreasing polarity solution (CH_3_OH/H_2_O); seven fractions were furnished (A–G) (Figure 1). Aliquot of fractions A (30 mg) and fraction B (30.8 mg) chromatographed on Silica gel TLC (1 mm) eluting with the organic phase of biphasic solution, CHCl3/MeOH/H_2_O (7:3:0.1), gave the pure compounds **1** (13.2 mg) and **2** (24.1 mg), respectively. The fraction C (137.6 mg) was purified by *flash*-CC SiO_2_ eluting with the lower phase of CHCl_3_/MeOH/H_2_O (13:7:2) biphasic solution to give another fraction (30.2 mg), which in turn was chromatographed through Silica gel TLC (1 mm), eluting with the organic phase of CHCl_3_/MeOH/H_2_O (13:7:2) solution and furnishing the pure compound **3** (11.3 mg). Fraction D yielded compound **4** (10 mg) in pure form. The fraction E (20 mg) was chromatographed on Silica gel TLC (0.5 mm) eluting with the lower phase of CHCl_3_/MeOH/H_2_O (13:7:2) solution, yielding the pure compounds **5** (8.1 mg) and **6** (4.5 mg). Fraction F represented pure compound **7** (29.5 mg). The fraction G (291.2 mg), chromatographed on RP-18 CC, eluting with decreasing polarity solution (CH_3_OH/H_2_O), furnished four fractions (G1-G4, Figure 1). The fraction G1 was purified on Silica gel TLC (1 mm) eluting with the organic phase of CHCl_3_/MeOH/H_2_O (13:7:2) solution, in turn, to give the pure compounds **8** (86.3 mg) and **9** (5.8 mg). The fraction G2 (37.3 mg), chromatographed on Silica gel TLC (1 mm) eluting with CHCl_3_/MeOH/H_2_O (14:6:1) solution, provided the pure compounds **10** (17 mg) and **11** (6.6 mg). The fraction G3 (21.2 mg) was purified on Silica gel TLC (0.5 mm), eluting with the lower phase of CHCl_3_/MeOH/H_2_O (13:7:4) solution, giving the pure compound **12** (7.3 mg). Finally, fraction G4 (52.3 mg) was chromatographed through Flash CC SiO_2_, eluting with the organic phase of CHCl*_3_*/MeOH/H_2_O (13:7:4) solution, yielding the pure compound **13** (1.6 mg).

### 4.8. Anti-HIV Enzymatic Assays

#### 4.8.1. Expression and Purification of Recombinant HIV-1RT

HIV-1 RT group M subtype B. Heterodimeric RT was expressed and purified essentially as previously [47] described. Briefly, protein was expressed in E. coli strain M15 containing the p6HRT-prot vector, induced with 1.7 mM isopropyl β-d-1-thiogalactopyranoside for 4 h. Protein purification was carried out with a BioLogic LP system (Biorad), using a combination of immobilized metal affinity and ion exchange chromatography. First, crude bacterial extract was clarified by centrifugation and loaded onto a Ni^2+^-NTA-Sepharose column pre-equilibrated with a loading buffer (50 mM sodium phosphate buffer pH 7.8, containing 0.3 M NaCl, 10% glycerol, and 10 mM imidazole); RT was eluted with an imidazole gradient in wash buffer (0–0.5 M). Fractions were collected and protein purity was checked by SDS-PAGE and found to be higher than 90%. The 1:1 ration between the p66/p51 subunits was also verified. Enzyme-containing fractions were pooled and diluted 1:1 with 50 mM sodium phosphate buffer pH 7.0, containing 10% glycerol, and then loaded into a Hi-trap heparin HP GE (Healthcare Lifescience) using a loading buffer (50 mM sodium phosphate buffer pH 7.0, containing 10% glycerol and 150 mM NaCl). RT was eluted with Elute Buffer 2 (50 mM Sodium Phosphate pH 7.0, 10% glycerol, 1 M NaCl). Fractions were collected and protein was dialyzed and stored in a buffer containing 50 mM Tris-HCl pH 7.0, 25 mM NaCl, 1 mM EDTA, and 50% glycerol. Catalytic activities and protein concentrations were determined. Enzyme-containing fractions were pooled and aliquots were stored at −80 °C.

#### 4.8.2. Expression and Purification of Recombinant HIV-1 IN and LEDGF

Recombinant 6xHis tagged IN protein was expressed and purified as described previously [48]. Briefly, IN was expressed in Escherichia coli strain BL21 (DE3). Initial purification was done using a Ni-Sepharose column with an imidazole gradient from 20 mM to 500 mM concentration in a 50 mM HEPES (pH 7.5) buffer containing 1 M NaCl, 7.5 mM CHAPS, and 2 mM β-mercaptoethanol. This was followed by a heparin column purification with a NaCl gradient from 0 to 1 M concentrations. His-LEDGF was purified by loading precipitate of cell lysate onto a heparin column and was eluted with an increasing NaCl gradient (200 mM to 1 M) in a 50 mM HEPES (pH 7.5) buffer containing 7.5 mM CHAPS and 2 mM β-mercaptoethanol. Peak fractions were pooled and loaded onto a Superdex 200 GL column and eluted in a buffer containing 50 mM HEPES (pH 7.5), 200 mM NaCl, and 2 mM β-mercaptoethanol. Fractions containing LEDGF were pooled and stored in 15% glycerol at −80 °C.

#### 4.8.3. RNase H Polymerase-Independent Cleavage Assay

HIV RT-associated RNase H activity was measured as described [49,50] (using the RNase H inhibitor RDS175911 as a control.) In a 100 µL reaction volume containing 50 mM Tris-HCl buffer with pH 7.8, 6 mM MgCl_2_, 1 mM dithiothreitol (DTT), 80 mM KCl, 0.25 µM hybrid RNA/DNA 5′-GAUCUGAGCCUGGGAGCU-Fluorescin-3′ (HPLC, dry, QC: Mass Check) (available from Metabion (Planegg, Germany)) 5′-Dabcyl-AGCTCCCAGGCTCAGATC-3′ (HPLC, dry, QC: Mass Check), there were increasing concentrations of an inhibitor, whose dilution was made in water, and 20 ng of wt RT, according to a linear range of dose-response curve. The reaction mixture was incubated for 1 h at 37 °C, stopped by the addition of EDTA, and products were measured with a multilabel counter plate reader Victor 3 (Perkin Elmer model 1420-051 (Waltham, MA, USA)) equipped with filters for 490/528 nm (excitation/emission wavelength).

#### 4.8.4. Homogeneous Time Resolved Fluorescence (HTRF) LEDGF Dependent Assay

The HTRF assay allows measuring the inhibition of 3′-processing and strand-transfer IN reactions in the presence of recombinant LEDGF/p75 cellular cofactor [43]. Briefly, 50 nM IN was pre-incubated with an increasing concentration of compounds for 1 h at room temperature in reaction buffer containing 20 mM HEPES pH 7.5, 1 mM DTT, 1% Glycerol, 20 mM MgCl2, 0.05% Brij-35, and 0.1 mg/mL BSA. To this mixture, 9 nM DNA donor substrate (5′-ACAGGCCTAGCACGCGTCG-Biotin-3′ annealed with 5′-CGACGCGTGGTAGGCCTGT-Biotin3′), 50 nM DNA acceptor substrate (5′-Cy5-ATGTGGAAAATCTCTAGCAGT-3′ annealed with 5′-Cy5- TGAGCTCGAGATTTTCCACAT-3′) and 50 nM LEDGF/p75 protein (or without LEDGF/p75 protein) were added and incubated at 37 °C for 90 min. After the incubation, 4 nM of Europium-Streptavidin were added to the reaction mixture, and the HTRF signal was recorded using a Perkin Elmer Victor 3 plate reader using a 314 nm for the excitation wavelength and 668 and 620 nm for the wavelength of the acceptor and the donor substrate emission, respectively.

## Figures and Tables

**Figure 1 molecules-26-04777-f001:**
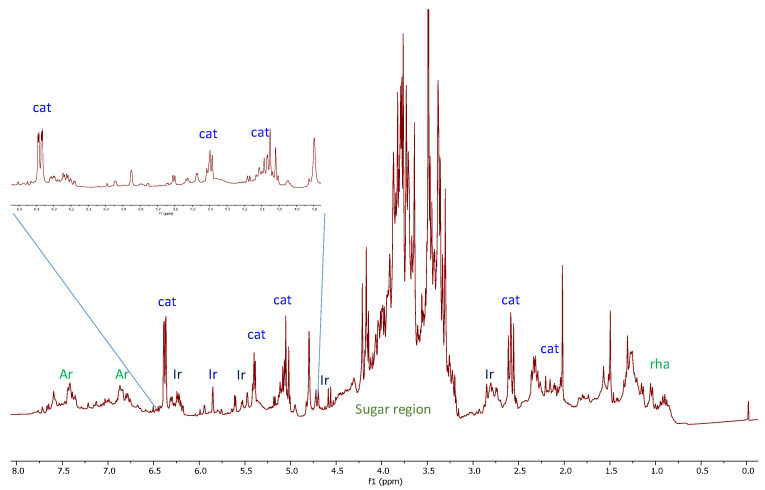
^1^H-NMR spectra of *S. trifoliata.* Spectra were acquired at 300 MHz, in 1:1 methanol-d4: phosphate buffer. Diagnostic signals of the main metabolites detected in extract are indicated by the following abbreviations: Ar = aromatic compounds, cat = catalpol (**1**), Ir = others iridoids, rha = rhamnose.

**Figure 2 molecules-26-04777-f002:**
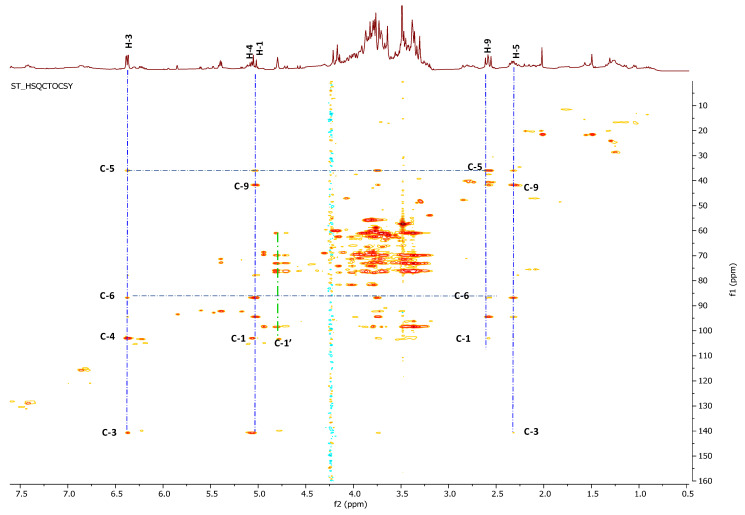
HSQC-TOCSY and ^1^H-NMR spectra of *S. trifoliata* crude extract.

**Figure 3 molecules-26-04777-f003:**
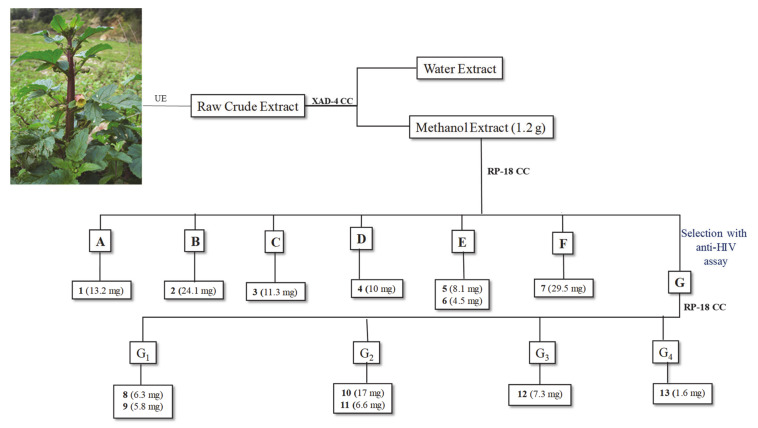
Fractionation scheme of *S. trifoliata* leaves hydroalcholic crude extract. CC = Column Chromatography; RP = reverse phase; UE = ultrasound extraction.

**Figure 4 molecules-26-04777-f004:**
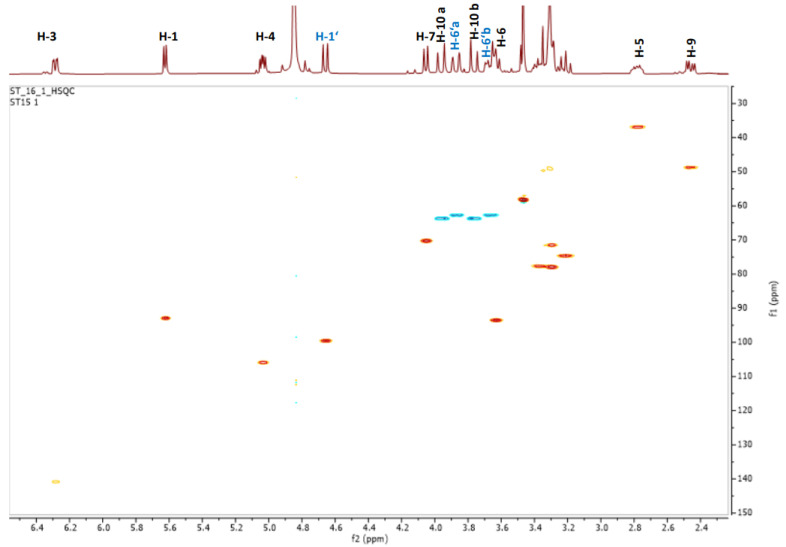
HSQC and ^1^H-NMR spectra of compound **5**.

**Figure 5 molecules-26-04777-f005:**
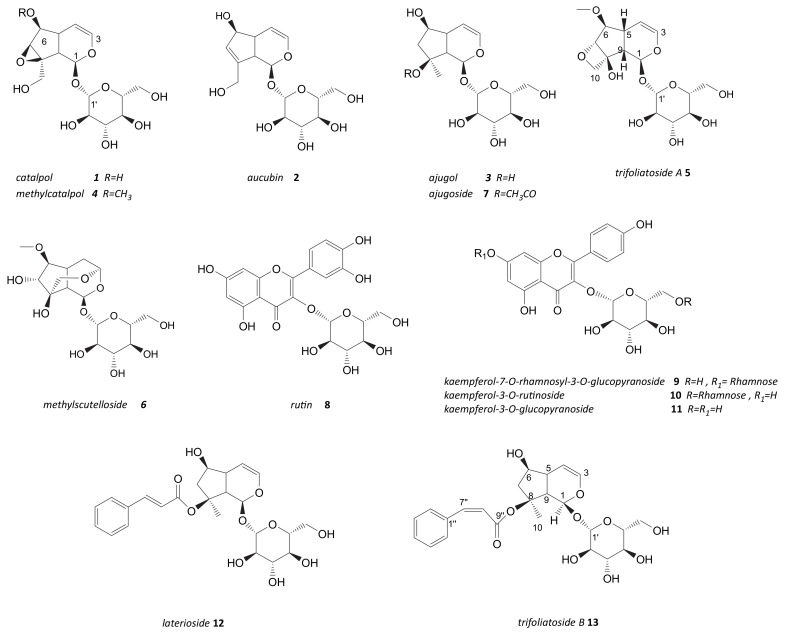
Chemical structure of iridoids (**1**–**7**, **12**–**13**) and flavonols glycosides (**8**–**11**) isolated from leaves of *S. trifoliata.*

**Figure 6 molecules-26-04777-f006:**
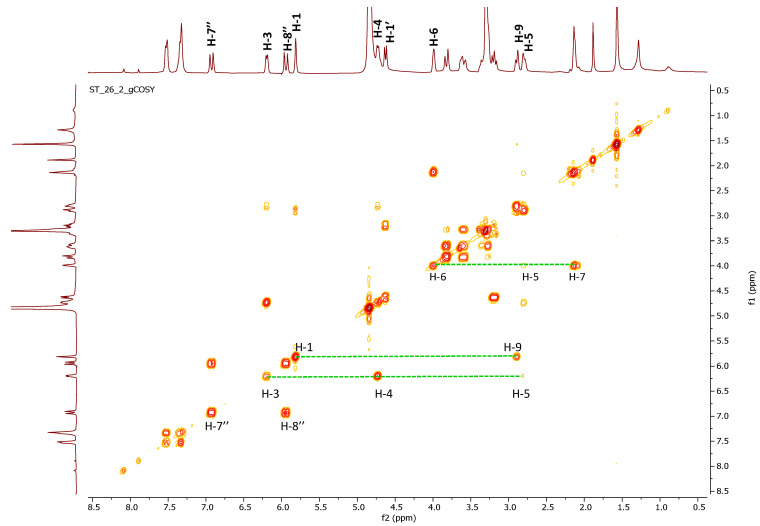
COSY and ^1^H-NMR spectra of compound **13**.

**Table 1 molecules-26-04777-t001:** NMR data of compound **5** in CD_3_OD.

Position	δ_C_ (Type)	δ_H_ (*J* in Hz)	HMBC (H→C)
1	93.0 (CH)	5.62 (d, *J* = 4.2 Hz)	C-3, C-5, C-1′
3	140.3 (CH)	6.28 (dd, J = 6.3, 1.8 Hz)	C-1, C-4
4	105.9 (CH)	5.03 (dd, *J* = 6.3, 3.6 Hz)	C-1, C-3, C-5, C-9
5	37.1 (CH)	2.77 (m)	C-6, C-8, C-9
6	93.5 (CH)	3.62 (d, *J* = 6 Hz)	C-5, C-7, C-OCH_3_
7	70.2 (CH)	4.04 (d, *J* = 7 Hz)	C-6, C-8,C-10
8	81.5 (C)		
9	48.9 (CH)	2.45 (dd, *J* = 4.2 Hz; 10.2 Hz)	C-4, C-5, C-10
10	63.7 (CH_2_)	3.95/ 3.75 (d, *J* = 12.0 Hz)	C-7, C-8, C-9
1′	99.6 (CH)	4.65 (d, *J* = 7.8 Hz)	C-1
2′	74.8 (CH)	3.20 (t, *J* = 7.8 Hz)	C-3′, C-1′
3′	77.8 (CH)	3.29 (ov)	C-4′
4′	71.6 (CH)	3.37 (ov)	C-3′, C-5′
5′	77.9 (CH)	3.30 (ov)	C-4′
6′	63.7 (CH_2_)	3.87(d, *J* = 13.8 Hz) 3.66 (ov)	C-5′
-OCH_3_	58.1 (CH_3_)	3.46 (s)	C-6

d = doublet, dd = doublet of doublets, m = multiplet; ov = overlapped; s = singlet; t = triplet; *J* values (Hz) are reported in brackets.

**Table 2 molecules-26-04777-t002:** Anti-HIV activity of crude extract, pure compounds, and fraction G.

	^a^ HIV-1 RNase H IC_50_ (µM)	^b^ HIV-1 IN LEDGF-Dependent IC_50_ (µM)
Crude extract	9.9 ± 0.93 ^c^	2.5 ± 0.4
1	>100 (100%) ^d^	>100 (100%) ^d^
2	>100 (100%) ^d^	>100 (100%) ^d^
3	>100 (100%) ^d^	>100 (77%) ^d^
4	>100 (100%) ^d^	>100 (80%) ^d^
5	>100 (93%) ^d^	95.5 ± 4.5
6	>100 (100%) ^d^	100.0 ± 0.5
7	>100 (100%) ^d^	74.0 ± 3.5
Fraction G	26.6 ± 1.3 ^c^	6.1 ± 0.92 ^c^
RDS 1759 ^e^	7.3 ± 0.10	
Raltegravir ^e^		0.058 ± 0.01

^a^ Compound concentration required to inhibit the HIV-1 RNase H activity by 50%. ^b^ Concentration required to inhibit the HIV-1 IN catalytic activities, in the presence of LEDGF, by 50%. ^c^ Extract concentration required to inhibit the HIV-1 RNase H activity by 50% expressed in µg/mL. ^d^ Percentage of control measured in the presence of 100 µM compound concentration. ^e^ Raltegravir and RDS1759 positive controls of IN LEDGF-dependent activity and RNase H function, respectively.

**Table 3 molecules-26-04777-t003:** Data ^1^H-NMR and ^13^C-NMR of compound **13** recorded in CD_3_OD.

Position	δ_C_ (Type)	δ_H_ (*J* in Hz)
1	94.1 (CH)	5.82 (brs)
3	141.5 (CH)	6.20 (dd, *J* = 6.3, 2.7 Hz)
4	103.2 (CH)	4.73 (m)
5	41.3 (CH)	2.80 (m)
6	76.8 (CH)	4.00 (m)
7	48.5 (CH_2_)	2.14 m
8	90.3 (C)	
9	49.4 (CH)	2.89 (m)
10	22.4 (CH_3_)	1.58 (s)
1′	99.8 (CH)	4.63 (d, *J* = 7.8 Hz)
2′	74.6 (CH)	3.20 (t, *J* = 7.8 Hz)
3′	77.8 (CH)	3.28 ov
4′	71.4 (CH)	3.28 ov
5′	77.9 (CH)	3.36 ov
6′	62.7 (CH_2_)	3.83 (d, *J* = 12.0 Hz) 3.60 m
1″	135.9 (C)	
2′	126.8 (CH)	7.39 ov
3″	121.9 (CH)	7.33 ov
4″	131.8 (CH)	7.58 ov
5″	126.8 (CH)	7.39 ov
6″	121.9 (CH)	7.39 ov
7″	142.4 (CH)	6.93 (d, *J* = 12.6 Hz)
8″	122.3 (CH)	5.95 (d, *J* = 12.6 Hz)
9″	168.2 (C)	

brs = broad singlet = doublet, m = multiplet; ov = overlapped; s = singlet; t = triplet, *J* values (Hz) are reported in brackets

**Table 4 molecules-26-04777-t004:** Anti-HIV activity of pure compounds isolated from fraction G.

Compounds	^a^ HIV-1 RNase H _(_µM)	^b^ HIV-1 IN LEDGF-Dependent (µM)
8	>100 (100%) ^c^	0.33 ± 0.04
9	>100 (100%) ^c^	0.11 ± 0.005
10	>100 (100%) ^c^	1.76 ± 0.017
11	>100 (100%) ^c^	0.024 ± 0.001
12	>100 (100%) ^c^	>100 (80%) ^c^
13	>100 (92%) ^c^	5.96 ± 0.10
Raltegravir ^d^		0.058 ± 0.010
RDS1759 ^d^	7.3 ± 0.10	

^a^ Compound concentration required to inhibit the HIV-1 RNase H activity by 50%. ^b^ Concentration required to inhibit the HIV-1 IN catalytic activities, in the presence of LEDGF, by 50%. ^c^ Percentage of control measured in the presence of 100 µM compound concentration. ^d^ Raltegravir and RDS1759 positive controls of IN LEDGF-dependent activity and RNase H function, respectively.

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
