# Peer review of "Chemical Characterization and Anti-HIV-1 Activity Assessment of Iridoids and Flavonols from *Scrophularia trifoliata"

_molecules, 2021, doi:10.3390/molecules26164777_

Round 1
Reviewer 1 Report
The MS represents a phytochemical analysis of alcoholic extracts of Scrophularia trifoliate, a plant characteristic for the Mediterranean region embracing Sardinia, Corsica and Gorgona Islands. The application of a series of very précised modern methods after the ultrasound assisted extraction procedure, namely, analytical TLC performed on Merck KIeselgel plates, NMR spectra determination, mass spectrometry via Maldi-Tof MS analyses, attained a very successful phytochemical analysis. The methodology for determining of the mechanism of anti-HIV activity involves anti-HIV enzymatic assays, expression and purification on recombinant HIV-1 IN and LEDGF, RNase H polymerase-independent cleavage assay and homogeneous time resolved fluorescence LEDGF dependent assay.
The study carried out using the mentioned methodology for phytochemical and virological analyses manifests successfully three flavonoids (rutin, kaempferol-7-O-rhamnosyl-3-O-glycopyranoside and kaempferol-3-O-glycopyranoside) as marked inhibitors of HIV-1.
Author Response
We are gratefuf to reviewer 1 for her positive comments
Reviewer 2 Report
Reviewer Comments on the Manuscript
Manuscript Number: Molecules 2021, 26, x. https://doi.org/10.3390/xxxxx
This manuscript describes “Chemical characterization and anti-HIV-1 activity assessment
of iridoids and flavonols from Scrophularia trifoliata’.
The authors showed significant efforts in search and characterization of novel antiviral agents against Human Immunodeficiency Virus type1 (HIV-1) from plants, Also, phytochemical investigation of Scrophularia trifoliata L. led them to isolate and characterize nine iridoid glycosides. Importantly, the two of them described for the first time, along with four flavonoid glycosides. In the present study, investigated, for the first time, the contents of a methanol extract of S. trifoliata leaves, in order to explore the potential antiviral activity against HIV-1. The antiviral activity has been evaluated in biochemical assays for the inhibition of HIV-1Reverse Transcriptase (RT)-associated Ribonuclease H (RNase H) activity and HIV-1 Integrase (IN).
The authors mentioned that the three isolated flavonoids, rutin, kaempferol-7-O-rhamnosyl-3-O-glucopyranoside and kaempferol-3-O-glucopyranoside inhibited specifically the HIV-1 IN activity at submicromolar concentration, with the latter being the most potent showing an IC50 value of 24 nM.
The compounds chemical characterization and antiviral properties was well described, although details that need to be corrected are listed below. It seems an important goal to discover plant based anti-HIV agents. Nevertheless, there are several minor problems with this paper that preclude its publication.
Minor concerns:
- In the abstract the compounds numbering and the significant biological activity of the few particular compounds need to be added. For example, the two newly discovered compounds need to be more explored in the abstract.
- In the results, coupling constant J should be italicized throughout the manuscript including Tables. Also “O” in the IUPAC names and Greek letters should be italicized.
- For better representation, the Figure 5. need to be modified with the addition of all compound’s names along with newly discovered compounds.
- Tables 2 and 4 legends carry the names of the standards used.
- In the Materials and Methods, section 4.6, all the compounds’ numbers should be in bold.

Author Response
In the abstract the compounds numbering and the significant biological activity of the few particular compounds need to be added. For example, the two newly discovered compounds need to be more explored in the abstract.
Answer The numbers of new compounds 5 and 13, as well as those of compounds with significant biological activity have been added
In the results, coupling constant J should be italicized throughout the manuscript including Tables. Also “O” in the IUPAC names and Greek letters should be italicized.
Answer done.
For better representation, the Figure 5. need to be modified with the addition of all compound’s names along with newly discovered compounds.
Answer The figure 5 has been modified according to reviewer suggestion.
Tables 2 and 4 legends carry the names of the standards used.
Answer The names of the standards used has been added in the legends of tables 2 and 4.
In the Materials and Methods, section 4.6, all the compounds’ numbers should be in bold.
Answer done